

# In silico study on anti-Chikungunya virus activity of hesperetin

Adrian Oo[1], Pouya Hassandarvish[1], Sek Peng Chin[2],
Vannajan Sanghiran Lee[2], Sazaly Abu Bakar[1] and Keivan Zandi[1]

[1] Tropical Infectious Disease Research and Education Centre, Department of Medical Microbiology Faculty of Medicine, University of Malaya, Kuala Lumpur, Malaysia
[2] Department of Chemistry, University of Malaya, Kuala Lumpur, Malaysia

## ABSTRACT

**Background:** The re-emerging, *Aedes spp.* transmitted Chikungunya virus (CHIKV) has recently caused large outbreaks in a wide geographical distribution of the world including countries in Europe and America. Though fatalities associated with this self-remitting disease were rarely reported, quality of patients' lives have been severely diminished by polyarthralgia recurrence. Neither effective antiviral treatment nor vaccines are available for CHIKV. Our previous in vitro screening showed that hesperetin, a bioflavonoid exhibits inhibitory effect on the virus intracellular replication. Here, we present a study using the computational approach to identify possible target proteins for future mechanistic studies of hesperetin.

**Methods:** 3D structures of CHIKV nsP2 (3TRK) and nsP3 (3GPG) were retrieved from Protein Data Bank (PDB), whereas nsP1, nsP4 and cellular factor SPK2 were modeled using Iterative Threading Assembly Refinement (I-TASSER) server based on respective amino acids sequence. We performed molecular docking on hesperetin against all four CHIKV non-structural proteins and SPK2. Proteins preparation and subsequent molecular docking were performed using Discovery Studio 2.5 and AutoDock Vina 1.5.6. The Lipinski's values of the ligand were computed and compared with the available data from PubChem. Two non-structural proteins with crystal structures 3GPG and 3TRK in complexed with hesperetin, demonstrated favorable free energy of binding from the docking study, were further explored using molecular dynamics (MD) simulations.

**Results:** We observed that hesperetin interacts with different types of proteins involving hydrogen bonds, pi-pi effects, pi-cation bonding and pi-sigma interactions with varying binding energies. Among all five tested proteins, our compound has the highest binding affinity with 3GPG at −8.5 kcal/mol. The ligand used in this study also matches the Lipinski's rule of five in addition to exhibiting closely similar properties with that of in PubChem. The docking simulation was performed to obtain a first guess of the binding structure of hesperetin complex and subsequently analysed by MD simulations to assess the reliability of the docking results. Root mean square deviation (RMSD) of the simulated systems from MD simulations indicated that the hesperetin complex remains stable within the simulation timescale.

**Discussion:** The ligand's tendencies of binding to the important proteins for CHIKV replication were consistent with our previous in vitro screening which showed its efficacy in blocking the virus intracellular replication. NsP3 serves as the highest potential target protein for the compound's inhibitory effect, while it is interesting to

Corresponding author
Keivan Zandi, keivan@um.edu.my

highlight the possibility of interrupting CHIKV replication via interaction with host cellular factor. By complying the Lipinski's rule of five, hesperetin exhibits drug-like properties which projects its potential as a therapeutic option for CHIKV infection.

## INTRODUCTION

Chikungunya virus (CHIKV) is an arthropod-borne virus classified under the arthritogenic group of *Alphavirus* genus from the *Togaviridae* family (*Assunção-Miranda, Cruz-Oliveira & Da Poian, 2013*). CHIKV is primarily transmitted by female mosquitoes of the *Aedes* genus, namely *Aedes aegypti* and *Aedes albopictus*, though *Culex ethiopicus* and *Anopheles stephensi* have also been reported as potential vectors. Over the years, massive outbreaks have been recorded in Africa, India and Indian Ocean islands as well as South and South-East Asia (*Beesoon et al., 2008*; *Josseran et al., 2006*; *Lanciotti et al., 2007*). In addition to the spread of CHIKV by *Aedes albopictus* to America and Europe, return of high viraemia infected travellers, have caused concerns of CHIKV transmittance to new endemic regions (*Angelini et al., 2007*; *Vega-Rúa et al., 2014*). Within 2–4 days following the initial mosquito bite, CHIKV enters the bloodstream where it disseminates to its respective target organs for subsequent replication. Infected patients generally suffer from symptoms such as polythermia (> 39 °C), maculopapular rashes, polyarthralgia, cephalgia and nauseas (*Taubitz et al., 2007*). Further complications involving the inflammation of heart muscles (myocarditis), brain and meninges (meningoencephalitis) as well as haemorrhage have also been reported but rare (*Farnon, Sejvar & Staples, 2008*). Though CHIKV is a self-limiting viral infection, 30–40% of patients have been complaining of recurrent joint pains which may last from a few weeks up to several years following recovery (*Manimunda et al., 2010*).

This small, spherical and enveloped virus contains a single positive-strand RNA genome which is about 11.8 kb in size (*Lo Presti et al., 2014*). The genome is made up of two open reading frames (ORFs), whereby the ORF covering 5′ two-thirds of the genome codes for the non-structural proteins, whereas CHIKV structural proteins are encoded by the 3′ ORF (*Snyder et al., 2013*). Cleavage of the polyproteins results in five structural (C, E3, E2, 6K, E1) and four nonstructural (nsP1, nsP2, nsP3, nsP4) proteins, with distinct functions.

While the structural proteins are mainly involved in the early stages of CHIKV replication cycle comprising of the attachment and entry of virus particles into infected cells, CHIKV nonstructural proteins are vital for the intracellular replication of new virus particles. The nonstructural proteins originated from a polyprotein (P1234) which is directly translated from the 5′ two-thirds of CHIKV genomic RNA. The section between nsP3 and nsP4 in the precursor protein is then cleaved in *cis*, resulting in
P123 and nsP4 proteins (*Merits et al., 2001*). Subsequently, an unstable replication complex made up of these proteins as well as cellular factors is assembled, which synthesizes minus-strand genomic RNA. Further cleavage of the P123 protein into nsP1 and P23, together with nsP4, gives rise to a replication complex within virus-induced cytopathic vacuoles (CPV I), capable of both the synthesis of sense and antisense genomic RNA (*Froshauer, Kartenbeck & Helenius, 1988*; *Salonen et al., 2003*; *Shirako & Strauss, 1994*). Following the complete processing into individual nsP1, nsP2, nsP3 and nsP4 proteins, a stable polymerase complex is formed capable of the production of the positive-sense genomic and subgenomic RNA molecules (*Lemm et al., 1994*).

The nsP1 protein via its guanine-7-methyltransferase and guanylyl transferase activities, catalyses the capping and methylation of newly synthesized viral mRNA in addition to its role in the synthesis of viral antisense RNA molecules (*Ahola & Kääriäinen, 1995*; *Mi & Stollar, 1991*). NsP1 is also involved in the attachment of virus polymerase complex to cellular membranes through interactions between the negatively-charged phospholipids with the amphipathic binding peptide (BP), and is enhanced by nsP1 palmitoylation (*Peränen et al., 1995*). Study has shown that alteration in the specific as well as the surrounding palmitoylation site reduces virus replication hence, its viability (*Ahola et al., 2000*). The nsP2 is multifunctional whereby it possesses the N-terminal RNA triphosphatase domain which catalyses the capping of viral RNA as well as the protein's RNA helicase activity (*Fros et al., 2013*; *Karpe, Aher & Lole, 2011*).

The C-terminal of the protein has cysteine protease activity which is essential for the autocatalysis of the P123 polyprotein, as well as a domain with methyltransferase-like activity of undiscovered function (*Bouraï et al., 2012*). It has also been reported that nsP2 regulates the transcription and translation of viral proteins using the host cell machinery besides protection against interferon's inhibitory activity (*Fros et al., 2010*). The nsP3 protein is made up of three domains, namely the evolutionarily highly conserved N-terminus, the centre domain unique only for *Alphaviruses* and the heavily phosphorylated C-terminus. Its macrodomain is suggested to mediate ADP-ribose 1″-phosphate and/or other ADP-ribose derivatives metabolism which regulate cellular activities (*Seyedi et al., 2016*). On the other hand, the C-terminus contains Src-homology 3 (SH3) binding motifs for the localization of cellular amphiphysins Amph1 and BIN1/Amph2 following virus infection which promote *Alphavirus* genome replication (*Neuvonen et al., 2011*). NsP3 also mediates cellular stress response by blocking the assembly of stress granules via the interaction between its SH3 binding motifs with Ras-GAP SH3 domain-binding protein (G3BP) (*Fros et al., 2012*). On the other hand, CHIKV nsP4 functions as RNA-dependent RNA polymerase vital for the elongation of the newly synthesised viral RNA molecules (*Rathore, Ng & Vasudevan, 2013*). A study also demonstrated the close interactions between CHIKV nsP4 as well as nsP3 proteins with the heat shock protein, HSP-90 which has been shown to play an important role in the replication process of several DNA and RNA viruses (*Rathore et al., 2014*).

Sphingosine kinases (SPKs) are lipid kinases involved in the conversion of sphingosine into sphingosine-1-phosphate (S1P). Two types of SPKs are present in mammalian cells, namely SPK1 and SPK2 which are translated from *SPHK1* gene on chromosome

17 and *SPHK2* gene on chromosome 19, respectively (*Neubauer & Pitson, 2013*). Previous studies have shown close association between SPK/S1P pathway and important cellular processes such as cell growth, inflammation and immune responses, with both types of SPKs having the opposing effects (*Maceyka et al., 2005*; *Okada et al., 2005*; *Olivera et al., 1999*; *Snider, Alexa Orr Gandy & Obeid, 2010*). Interestingly, in addition to its contribution towards the development of degenerative diseases in humans, SPKs also play important roles in several viral infections, including Dengue and Influenza A viruses for SPK1 as well as Hepatitis C virus, Kaposi's sarcoma-associated herpes virus, and most importantly for our study, CHIKV for SPK2 (*Clarke et al., 2016*; *Couttas et al., 2014*; *Dai et al., 2014*; *Reid et al., 2015*; *Seo et al., 2013*; *Yamane et al., 2014*).

Flavonoids are a group of polyphenolic plant compounds which are synthesised via the phenylpropanoid pathway. Over the years, different types of flavonoids have been the subject of research for their wide range of medicinal benefits–anti-oxidant, anti-tumour, anti-inflammation and anti-microbial activities (*Martínez-Pérez et al., 2016*; *Paul Dzoyem et al., 2013*; *Procházková, Boušová & Wilhelmová, 2011*; *Serafini, Peluso & Raguzzini, 2010*). Indeed, various flavonoids have been reported for their antiviral properties against Dengue Virus (DENV), Herpes Simplex Virus (HSV), Human Cytomegalovirus (HCMV) and etc (*Cotin et al., 2012*; *Lyu, Rhim & Park, 2005*; *Moghaddam et al., 2014*; *Ono et al., 1989*; *Zandi et al., 2011*). In our previous study, we have found that hesperetin which belongs to the flavanone class of flavonoids possesses effective inhibitory effect on the in vitro intracellular replication of CHIKV (*Ahmadi et al., 2016*). Hence, at present we will be performing an in silico investigation to identify the possible target proteins for subsequent mechanistic study of hesperetin on CHIKV.

## MATERIALS AND METHODS

### Receptor and ligand preparation

The receptors used in this study are the three-dimensional protein structures of CHIKV nsP1, nsP2, nsP3, nsP4 and the mammalian cells' SPK2. CHIKV nsP1 and nsP4 as well as the SPK2 protein structures were modelled using the Iterative Threading Assembly Refinement (I-TASSER) server based on the amino acid sequences obtained from Universal Protein Resource (UniProt) (*Roy, Kucukural & Zhang, 2010*; *Yang et al., 2015*; *Zhang, 2008*). On the other hand, nsP2 and nsP3 crystal structures were retrieved from the Protein Data Bank (PDB) with PDB IDs of 3TRK and 3GPG respectively. Using the Discovery Studio 2.5 software, water molecules were removed and CHARMM27 force field was applied onto these protein structures. Subsequently, steric overlaps were removed and the structures were processed 1,000 steps using the smart-minimizer algorithm. Hesperetin was used as the ligand in this experiment, of which the ligand structure was generated using ChemDraw software (CambridgeSoft). The 2D-structure was then minimized using the Discovery Studio 2.5 software followed by the preparation of both the ligand and receptors during which hydrogen molecules were added using the AutoDock Vina 1.5.6 software, which were then saved as PDBQT files.

**Table 1 Parameters used for molecular docking of hesperetin with each protein of interest.** All grid boxes with spacing size of 1.000 Å have sufficient sizes to cover the entire protein structures during molecular docking.

| Proteins | Center-X | Center-Y | Center-Z | Size-X | Size-Y | Size-Z |
|----------|----------|----------|----------|--------|--------|--------|
| Nsp1 | 8.490 | −21.201 | −6.330 | 96 | 78 | 70 |
| 3TRK | 11.569 | 24.420 | 21.707 | 52 | 76 | 56 |
| 3GPG | 8.490 | −21.201 | −21.201 | 96 | 78 | 70 |
| NsP4 | 76.026 | 77.432 | 77.204 | 78 | 84 | 58 |
| SPK2 | 67.704 | 77.970 | 71.232 | 72 | 72 | 68 |

## Molecular docking

In our study, the binding properties of the ligand to each protein were analysed using the AutoDock Vina 1.5.6 software. Blind dockings were performed in which grid boxes of sizes sufficient to cover the entire receptors of interest were constructed according to the parameters as tabulated in Table 1. All grid boxes were formed at spacing size of 1.000 Å. Following the docking procedure, the resulting PDBQT output file was first opened in the PyMOL software in which all protein conformations were converged in a single file for further analysis. Each conformation was then analysed using the AutoDock Vina 1.5.6 and Discovery Studio 2.5 software whereby information such as binding affinities, interaction energies, van der Waals energies, electrostatic energies, hydrogen bonding, pi-pi interactions, pi-cation interactions and close contacting residues were obtained and recorded. Separately, the Lipinski's values for hesperetin were determined using the Lipinski Filters (SCFBio, Delhi) (*Jayaram et al., 2012*; *Lipinski, 2004*).

## Molecular dynamics simulations

The docked complexes of hesperetin to the non-structural proteins–3GPG and 3TRK which demonstrated favourable free energy of binding during molecular docking were used as starting structures for the molecular dynamics (MD) simulations. A total of 10 ns MD simulations were performed using AMBER 14 (*Case et al., 2010*). Amber ff14SB force field (*Maier et al., 2015*) was used to describe the proteins, waters and counter-ions. Antechamber was used to determine GAFF atom types for hesperetin, following Amber standard protocol. Restrained ElectroStatic Potential (RESP) charges were employed for ligands (*Bayly et al., 1993*). MD simulations were performed using constant pressure (P) and temperature (T), NPT ensemble, maintaining the P and T at 1.0 atm and 310 K, respectively, by means of anisotropic P scaling and Langevin dynamics. The periodic boundary conditions based on the particle mesh Ewald method with a non-bonded cutoff of 8 Å were used. The integration time step was set at 2 fs and the SHAKE algorithm was used to constrain bond lengths involving hydrogen atoms. The system simulated underwent two stages of minimization using steepest descent and conjugate gradient with different parts of the system gradually released in stages. Then, it was slowly heated from 0 to 310 K within 500 ps with restraints on the complex. After the heating stage, the system was equilibrated for 1 ns, followed by another 10 ns of a relaxed MD run. The complete system trajectory was collected every 2 ps for analysis. Trajectories analyses were performed using the CPPTRAJ modules

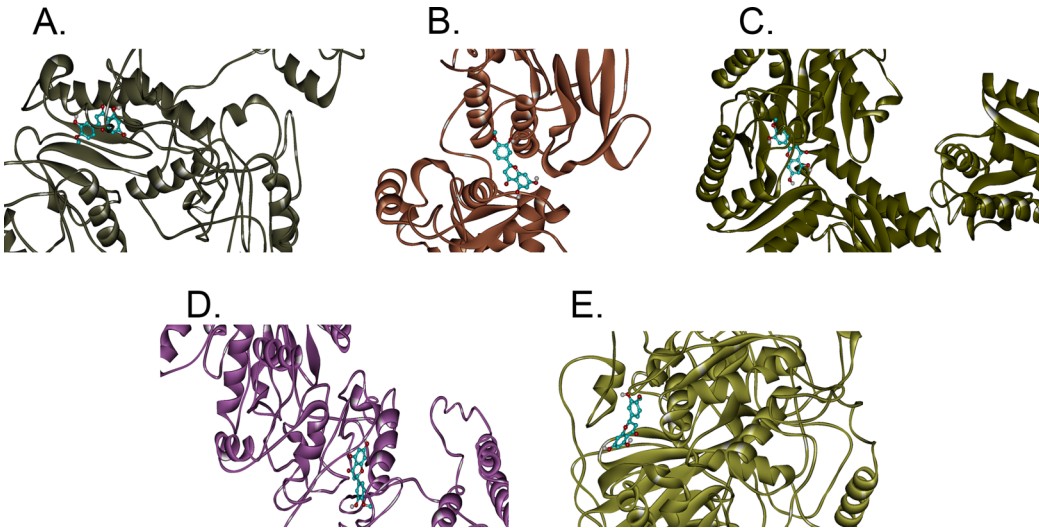

**Figure 1 The binding positions with the highest binding affinities of hesperetin (yellow ball and stick structure) when docked against different proteins (flat ribbon coloured from N-to-C terminal).** (A) nsP1 with binding affinity of −7.6 kcal/mol, (B) 3TRK with binding affinity of −6.9 kcal/mol, (C) 3GPG with binding affinity of −8.5 kcal/mol, (D) nsP4 with binding affinity of −7.7 kcal/mol, (E) SPK2 with binding affinity of −7.7 kcal/mol.

(*Roe & Cheatham, 2013*) of AMBER 16. The relative binding free energy for the complex was evaluated using MM-GBSA approach as implemented in Amber, using MMPBSA.py module (*Miller et al., 2012*) in Amber 16. A total of 500 snapshots from the relax MD trajectories were included in the calculations.

## RESULTS

In this study, hesperetin was screened for potential inhibitory activities against all four CHIKV's non-structural proteins, namely nsP1, nsP2, nsP3 and nsP4 in addition to SPK2 via the docking of the compound to each protein. Figures 1A–1E show the bindings of hesperetin with the highest binding affinities upon docking against the proteins of interest. As illustrated, hesperetin exhibits the strongest binding affinity against 3GPG (−8.5 kcal/mol), followed by nsP4 and SPK2 (−7.7 kcal/mol), nsP1 (−7.6 kcal/mol) and finally 3TRK (−6.9 kcal/mol). The total interaction energies between the ligand and proteins resulting from the electrostatic as well as the weaker van der Waals forces are as tabulated in Table 2.

Besides looking into the binding affinities and energies involved in the interaction of hesperetin with all five proteins, we have also taken into record of the bonds formed between the ligand and the binding sites on each protein. For nsP1, hydrogen bonds observed between this ligand-receptor proteins interactions are recorded in Table 3. Hesperetin exhibits pi-pi and pi-cation interactions with TYR185, PHE43 and ARG70, ARG252, LYS442, respectively (Table 4). The Discovery Studio 2.5 shows close amino acid residues found in the binding pocket between hesperetin and nsP1, namely LEU15, ALA17, LEU18, GLN19, ALA21, TYR22, ASN39, ALA40, PHE43, ALA40, PRO68, ALA69, ARG70, ARG71 and ALA104 as displayed in Fig. 2.

**Table 2 Binding energies between hesperetin and different types of proteins.** The energies involved in each ligand-receptor interaction was analysed using the Discovery Studio 2.5.

| Proteins | Electrostatic interaction energy (kcal/mol) | Van der Waals interaction energy (kcal/mol) | Interaction energy (kcal/mol) |
|---|---|---|---|
| NsP1 | 16.785 | −29.699 | −12.914 |
| 3TRK | −454.598 | −83.186 | −537.784 |
| 3GPG | −405.273 | −129.780 | −535.053 |
| NsP4 | −32.689 | −28.904 | −61.594 |
| SPK2 | 14.815 | −27.477 | −12.663 |

**Table 3 Hydrogen bonding between hesperetin and CHIKV nsP1.** This table documents the residues involved in the hydrogen bond formation as well as the angle DHA and length of the hydrogen bonds as analysed using the Discovery Studio 2.5. The binding affinities as ranked by the AutoDock Vina 1.5.6 are recorded in the final column of the table.

| Hydrogen bonds | Angle DHA (°) | Distance (Å) | Binding affinity (kcal/mol) |
|---|---|---|---|
| Hesperetin:UNK1:H2–nsP1:ASP310:OD1 | 167.07° | 1.66709 | −7.6 |
| nsP1:GLY311:HN–Hesperetin:UNK1:O19 | 159.57° | 2.13746 | |
| nsP1:ARG365:HH21–Hesperetin:UNK1:O20 | 93.9791° | 2.49524 | |
| nsP1:ARG365:HH22–Hesperetin:UNK1:O20 | 104.508° | 2.33411 | |
| nsP1:ARG365:HH22–Hesperetin:UNK1:O21 | 121.423° | 2.29742 | |
| Hesperetin:UNK1:H1–nsP1:HIS45:ND1 | 149.472° | 2.28058 | −7.5 |
| nsP1:GLY190:HN–Hesperetin:UNK1:O20 | 124.29° | 2.43961 | |
| Hesperetin:UNK1:H1–nsP1:ASP36:OD2 | 125.798° | 2.08242 | −7.4 |
| Hesperetin:UNK1:H2–nsP1:PRO68:O | 146.956° | 2.24366 | |
| nsP1:ARG71:HH11–Hesperetin:UNK1:O20 | 158.052° | 2.07172 | |
| Hesperetin:UNK1:H3–nsP1:GLU129:O | 119.206° | 2.47249 | |
| nsP1:LEU108:HN–Hesperetin:UNK1:O20 | 153.395° | 2.33121 | |
| nsP1:LEU135:HN–Hesperetin:UNK1:O17 | 125.791° | 2.08609 | |
| nsP1:ARG70:HN–Hesperetin:UNK1:O17 | 134.179° | 2.09364 | −7.3 |
| nsP1:ARG252:HH12–Hesperetin:UNK1:O20 | 171.884° | 2.30037 | |
| Hesperetin:UNK1:H1–nsP1:ASP448:OD1 | 154.403° | 1.82012 | −7.2 |
| Hesperetin:UNK1:H3–nsP1:VAL475:O | 159.829° | 1.84851 | |
| Hesperetin:UNK1:H3–nsP1:SER44:OG | 119.912° | 2.48095 | −6.9 |
| nsP1:ARG71:HH12–Hesperetin:UNK1:O20 | 126.228° | 2.28728 | |

CHIKV's nsP2 is an important protein for the initial stage of new viral RNA synthesis. In this study we carried out molecular docking of hesperetin on the cysteine protease domain of the virus nsP2, as denoted as 3TRK. The hydrogen bonds between hesperetin and 3TRK are as shown in Table 5. Pi-effects are observed at tyrosine residues of TYR1079 and TYR1177 as well as the phenylalanine PHE1225 (Table 4). In Fig. 3, we have demonstrated the close residues of 3TRK with hesperetin at ALA1046, TYR1047, SER1048, TYR1079, ASN1082, LEU1206, GLN1241 and GLY1245.

Another protein studied is the nsP3 protein as denoted 3GPG, which plays multiple roles in the virus life cycle. Table 6 documents the hydrogen bonds linking hesperetin and

**Table 4 Pi-interactions between hesperetin and respective proteins.** The residues involved and length of each pi-interaction as well as the types of pi-interactions formed were analysed using the Discovery Studio 2.5.

| Protein | Binding | Distance (Å) | Interaction |
|---|---|---|---|
| NsP1 | Hesperetin:UNK1–ARG70:NE | 5.2141 | Pi-cation |
| | Hesperetin:UNK1–nsP1:TYR185 | 6.18796 | Pi-pi |
| | Hesperetin:UNK1–nsP1:ARG252:NE | 6.85892 | Pi-cation |
| | Hesperetin:UNK1–ARG252:NE | 6.0649 | Pi-cation |
| | Hesperetin:UNK1–nsP1:PHE43 | 6.334 | Pi-pi |
| | Hesperetin:UNK1–nsP1:LYS442:NZ | 5.9124 | Pi-cation |
| 3TRK | Hesperetin:UNK1–3TRK:TYR1079 | 4.72556 | Pi-pi |
| | Hesperetin:UNK1–3TRK:TYR1079 | 5.65846 | Pi-pi |
| | Hesperetin:UNK1–3TRK:TYR1177 | 4.36635 | Pi-pi |
| | Hesperetin:UNK1–3TRK:PHE1225:HB2 | 2.74925 | Pi-sigma |
| 3GPG | Hesperetin: UNK1–3GPG:TRP148 | 5.10212 | Pi-pi |
| | Hesperetin: UNK1–3GPG:TRP148 | 4.15778 | Pi-pi |
| | Hesperetin:UNK1–3GPG:LYS118:NZ | 3.86706 | Pi-cation |
| | Hesperetin:UNK1–3GPG:GLY30:HA1 | 2.84749 | Pi-sigma |
| | Hesperetin:UNK1–3GPG:VAL113:HB | 2.9202 | Pi-sigma |
| | Hesperetin:UNK1–3GPG:TYR63 | 4.02708 | Pi-pi |
| | 3GPG:TRP148–Hesperetin:UNK1 | 4.22731 | Pi-pi |
| | 3GPG:TRP148–Hesperetin:UNK1 | 4.05885 | Pi-pi |
| NsP4 | Hesperetin:UNK1–nsP4:ARG374:NE | 5.53545 | Pi-cation |
| | Hesperetin:UNK1–nsP4:ARG70:NE | 5.3946 | Pi-cation |
| | Hesperetin:UNK1–nsP4:ARG70:NE | 6.64674 | Pi-cation |
| | Hesperetin:UNK1–nsP4:ARG70:NE | 5.65215 | Pi-cation |
| | Hesperetin:UNK1–nsP4:ARG70:NE | 5.28131 | Pi-cation |
| | Hesperetin:UNK1–nsP4:TYR22 | 4.15499 | Pi-pi |
| | Hesperetin:UNK1–nsP4:ARG70:NE | 4.49028 | Pi-cation |
| | Hesperetin:UNK1–nsP4:TYR185 | 6.16947 | Pi-pi |
| | Hesperetin:UNK1–nsP4:ARG70:NE | 5.20374 | Pi-cation |
| | Heesperetin:UNK1–nsP4:ARG70:NE | 4.51522 | Pi-cation |
| | Hesperetin:UNK1–nsP4:ARG70:HD1 | 2.61933 | Pi-sigma |
| SPK2 | Hesperetin:UNK1–SPK2:TYR372 | 4.91884 | Pi-pi |
| | SPK2:TYR372–Hesperetin:UNK1 | 4.34443 | Pi-pi |
| | Hesperetin:UNK1–SPK2:ARG351:NE | 6.2923 | Pi-cation |

3GPG obtained from our docking experiment. Interestingly, we also observed pi-pi interactions with TRP148 and TYR63, pi-cation interaction with LYS118 as well as pi-sigma with GLY30 and VAL113 (Table 4). Amino acid residues ALA22, ALA23, ASN24, ASP31, GLY32, VAL33, CYS34, PRO107, LEU109, THR111, GLY112, VAL113, TYR114, CYS143, ARG144 and ASP145 were also found closely in the binding pocket with hesperetin (Fig. 4).

We studied the potential inhibitory activity of hesperetin against CHIKV's nsP4 which serve as the polymerase for the elongation of newly synthesised RNA strands by

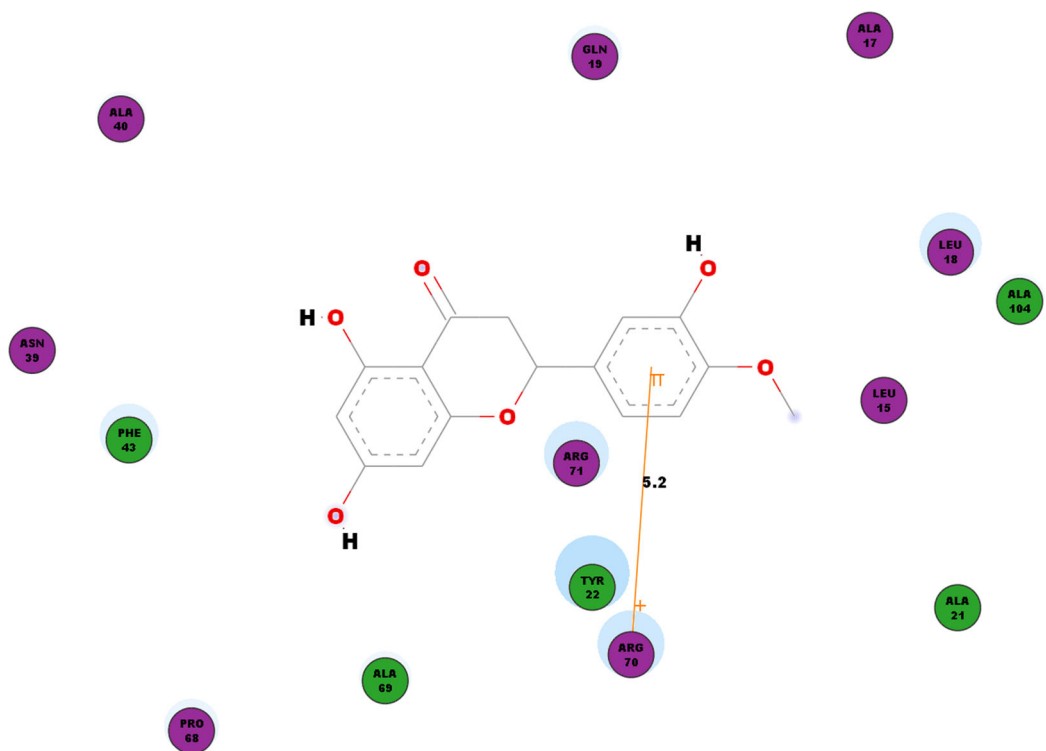

**Figure 2 2D diagram of interaction between hesperetin and nsP1.** The diagram shows the ligand-receptor interactions and close amino acid residues found in the binding pocket.

**Table 5 Hydrogen bonding between hesperetin and 3TRK.** This table documents the residues involved in the hydrogen bond formation as well as the angle DHA and length of the hydrogen bonds as analysed using the Discovery Studio 2.5. The binding affinities as ranked by the AutoDock Vina 1.5.6 are recorded in the final column of the table.

| Hydrogen bonds | Angle DHA (°) | Distance (Å) | Binding affinity (kcal/mol) |
| --- | --- | --- | --- |
| Hesperetin:UNK1:H3–3TRK:ASP1246:OD1 | 132.214 | 2.19274 | −6.9 |
| 3TRK:TRP1084:HE1–Hesperetin:UNK1:O18 | 140.389 | 2.20564 | |
| 3TRK:TRP1084:HE1–Hesperetin:UNK1:O19 | 148.713 | 2.13609 | |
| Hesperetin:UNK1:H3–3TRK:Ser1048:OG | 144.592 | 2.32087 | −6.8 |
| Hesperetin:UNK1:H1–3TRK:GLN1241:O | 93.425 | 2.48789 | −6.6 |
| 3TRK:ALA1180:HN–Hesperetin:UNK1:O20 | 136.259 | 2.09817 | −6.5 |
| 3TRK:LYS1091:HZ2–Hesperetin:UNK1:O18 | 134.881 | 2.44139 | −6.4 |
| 3TRK:ARG1267:HH12–Hesperetin:UNK1:O19 | 141.467 | 2.27262 | |
| 3TRK:ARG1267:HH22–Hesperetin:UNK1:O19 | 149.413 | 2.06422 | |
| Hesperetin:UNK1:H1–3TRK:LEU1065:O | 144.638 | 2.23416 | |
| 3TRK:PHE1225:HN–Hesperetin:UNK1:O10 | 145.498 | 2.33505 | −6.1 |
| 3TRK:ASN1135:HN–Hesperetin:UNK1:O18 | 126.233 | 2.487381 | |
| 3TRK:ASN1135:HN–Hesperetin:UNK1:O19 | 129.610 | 2.38619 | |

simulating the docking process of the compound against this protein. In addition to the hydrogen bonds as shown in Table 7, most pi interactions formed in this ligand-protein pair are pi-cations at ARG374 and ARG70 (Table 4). The remaining pi-pi interactions

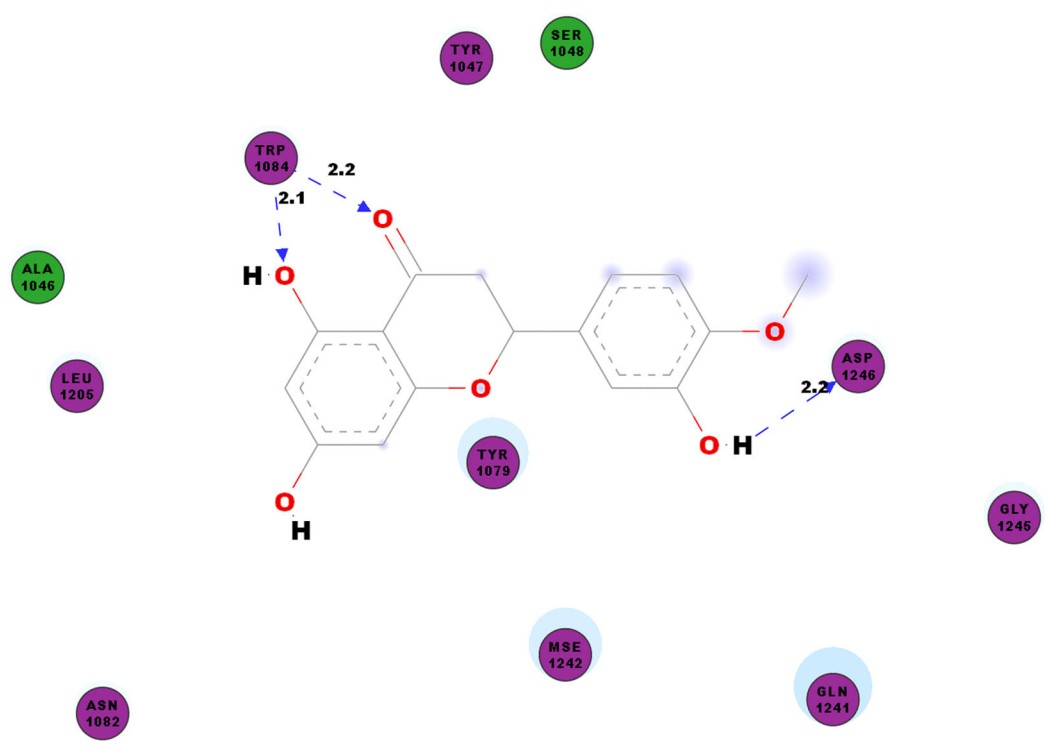

**Figure 3 2D diagram of interaction between hesperetin and 3TRK.** The diagram demonstrates the ligand-receptor interactions and close amino acid residues found in the binding pocket.

**Table 6 Hydrogen bonding between hesperetin and 3GPG.** This table documents the residues involved in the hydrogen bond formation as well as the angle DHA and length of the hydrogen bonds as analysed using the Discovery Studio 2.5. The binding affinities as ranked by the AutoDock Vina 1.5.6 are recorded in the final column of the table.

| Hydrogen bonds | Angle DHA (°) | Distance (Å) | Binding affinity (kcal/mol) |
|---|---|---|---|
| Hesperetin:UNK1:H1–3GPG:TYR142:O | 123.476 | 1.97816 | −8.5 |
| 3GPG:LEU108:HN–Hesperetin:UNK1:O17 | 162.843 | 2.17416 | |
| 3GPG:SER110:HN–Hesperetin:UNK1:O20 | 145.746 | 2.0736 | |
| 3GPG:VAL33:HN–Hesperetin:UNK1:O20 | 125.205 | 2.35031 | −8.3 |
| 3GPG:ARG144:HN–Hesperetin:UNK1:O19 | 159.099 | 1.95421 | |
| Hesperetin:UNK1:H2–3GPG:GLU17:O | 165.815 | 1.93615 | −7.3 |
| 3GPG:THR122:HN–Hesperetin:UNK1:O | 145.209 | 2.13631 | |
| 3GPG:ASN72:HD22–Hesperetin:UNK1:O20 | 139.167 | 2.03974 | |
| 3GPG:SER110:HN–Hepseretin:UNK1:O21 | 152.590 | 1.95709 | −7.0 |
| 3GPG:TRP41:HE1–Hesperetin:UNK1:O18 | 123.985 | 2.27795 | |
| 3GPG:LYS7:HZ2–Hesperetin:UNK1:O20 | 144.885 | 2.15643 | |
| Hesperetin:UNK1:H3–3GPG:TYR4:O | 148.477 | 2.16364 | −6.9 |
| 3GPG:GLN157:HE22–Hesperetin:UNK1:O20 | 129.097 | 2.30335 | |
| Hesperetin:UNK1:H1–3GPG:GLY116:O | 128.866 | 2.03758 | |
| Hesperetin:UNK1:H2–3GPG:GLU17:OE2 | 128.230 | 2.31651 | |
| 3GPG:ARG120:HH21–Hesperetin:UNK1:O17 | 123.759 | 2.34233 | |
| 3GPG:LEU108:HN–Hesperetin:UNK1:O17 | 178.542 | 1.72039 | |

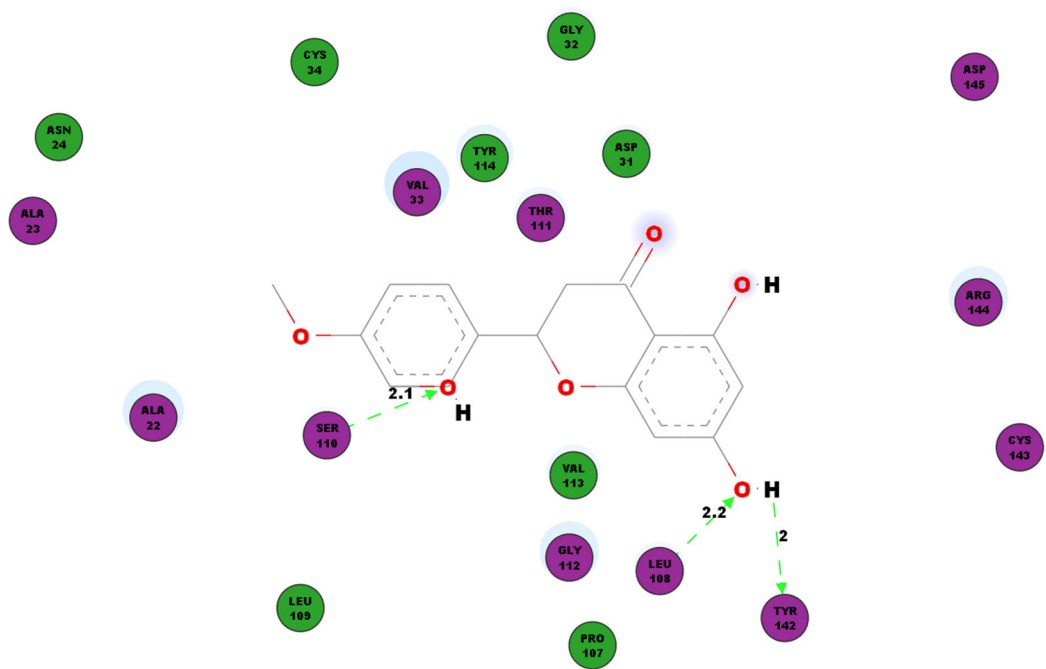

**Figure 4  2D diagram of interaction between hesperetin and 3GPG.** The diagram illustrates the ligand-receptor interactions and close amino acid residues found within the binding pocket.

formed at TYR22 and TYR185, whereas pi-sigma interaction was observed at ARG70 (Table 4). Illustrated in Fig. 5 are the close residues including TRP199, ALA200, LEU313, CYS315, LYS316, Val321, VAL326, PHE328, ASN363, ILE366, THR372, GLN373.

In addition to the virus proteins, we have also docked our compound of interest to cellular factor vital for virus replication such as SPK2. Our work shows hydrogen bonds between hesperetin and SPK2 as tabulated in Table 8. TYR372 and ARG351 interact with the ligand via pi-pi and pi-cation bonding respectively (Table 4). The hesperetin-SPK2 interaction also saw amino acid residues such as HIS251, ASN255, ASP259, CYS329, LEU425, PRO426, LEU427, PRO428, LEU432, ALA433, GLU624, GLN625, VAL626, GLU627 and TYR628 in close vicinities within the binding pocket (Fig. 6).

In order to ensure that the designed hesperetin molecule used in this experiment possesses the necessary drug-like properties to be developed as a therapeutic option for CHIKV infections, the Lipinski's values of our compound were determined and compared with the available data in PubChem (Table 9).

Hesperetin was found to bind and remain in the binding site throughout the simulation (see Movie S1). For both systems (3GPG and 3TRK), root mean square deviation (RMSD) computed using the backbone atoms of the complex with respect to the minimized starting structure was found to be less than 1.8 Å (Fig. 7). This showed that the systems are well equilibrated and do not deviate greatly from their initial starting structure. The lowest energy structures of the complexes are shown in Fig. 8. To explore the binding of hesperetin to 3GPG and 3TRK, the relative free energy of binding was computed from enthalpy contributions, using snapshots extracted from

**Table 7 Hydrogen bonding between hesperetin and CHIKV nsP4.** This table documents the residues involved in the hydrogen bond formation as well as the angle DHA and length of the hydrogen bonds as analysed using the Discovery Studio 2.5. The binding affinities as ranked by the AutoDock Vina 1.5.6 are recorded in the final column of the table.

| Hydrogen bonds | Angle DHA (°) | Distance (Å) | Binding affinity (kcal/mol) |
|---|---|---|---|
| nsP4:GLN364:HE22–Hesperetin:UNK1:O21 | 148.552 | 2.20397 | −7.7 |
| nsP4:ARG374:HH21–Hesperetin:UNK1:O19 | 121.976 | 2.4414 | |
| Hesperetin:UNK1:H3–nsP4:ALA104:O | 127.732 | 2.19555 | −7.6 |
| Hesperetin:UNK1:H1–nsP4:ALA103:O | 147.014 | 1.88423 | −7.5 |
| nsP4:GLY190:HN–Hesperetin:UNK1:O20 | 123.296 | 2.41584 | −7.4 |
| Hesperetin:UNK1:H2–nsP4:ARG110:O | 117.610 | 2.16176 | |
| nsP4:LEU108:HN–Hesperetin:UNK1:O20 | 147.895 | 2.27281 | |
| nsP4:LEU135:HN–Hesperetin:UNK1:O17 | 126.767 | 2.09964 | |
| Hesperetin:UNK1:H1–nsP4:THR437:O | 111.885 | 2.2692 | |
| Hesperetin:UNK1:H2–nsP4:ASP479:OD1 | 108.572 | 2.17349 | |
| nsP4:THR437:HN–Hesperetin:UNK1:O17 | 174.240 | 2.08043 | |
| nsP4:LEU180:HN–Hesperetin:UNK1:O19 | 117.866 | 2.47914 | |

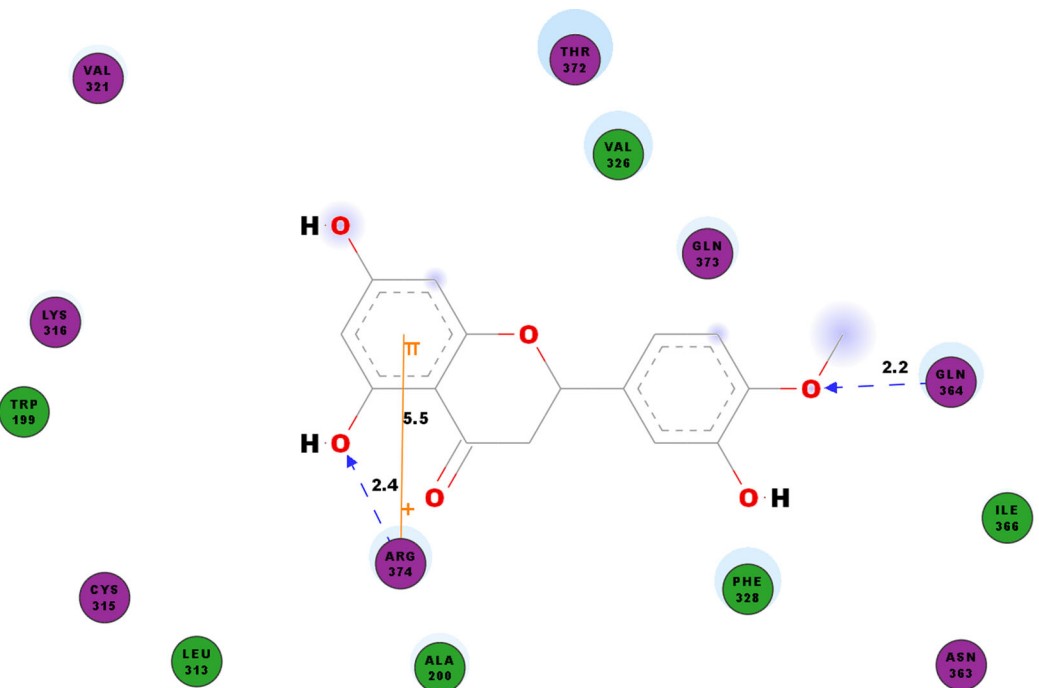

**Figure 5 2D diagram of interaction between hesperetin and nsP4.** The diagram shows the ligand-receptor interactions and close amino acid residues found in the binding pocket.

the trajectories, following the MM-GBSA approach (*Kollman et al., 2000*; *Srinivasan et al., 1998*). Table 10 lists the contributions to the binding free energy for the simulated complexes–favourable contributions to the binding arose from van der Waals interactions and the non-polar part of the solvation free energy, as opposed to

**Table 8 Hydrogen bonding between hesperetin and SPK2.** This table documents the residues involved in the hydrogen bond formation as well as the angle DHA and length of the hydrogen bonds as analysed using the Discovery Studio 2.5. The binding affinities as ranked by the AutoDock Vina 1.5.6 are recorded in the final column of the table.

| Hydrogen bonds | Angle DHA (°) | Distance (Å) | Binding affinity (kcal/mol) |
|---|---|---|---|
| SPK2:ARG227:HE–Hesperetin:UNK1:O21 | 132.729 | 2.39668 | −7.7 |
| Hesperetin:UNK1:H3–SPK2:ALA431:O | 106.199 | 2.32194 | |
| SPK2:GLN223:HE–Hesperetin:UNK1:O19 | 111.042 | 2.45074 | −7.6 |
| SPK2: GLN625:HN–Hesperetin:UNK1:O17 | 110.511 | 2.45589 | |
| Hesperetin:UNK1:H1–SPK2:GLU624:OE1 | 139.804 | 2.27908 | |
| Hesperetin:UNK1:H2–SPK2:ARG373:O | 129.174 | 1.92199 | |
| Hesperetin:UNK1:H3–SPK2:ASP342:OD1 | 141.848 | 2.21488 | |
| SPK2:ARG227:HE–Hesperetin:UNK1:O21 | 130.540 | 2.3656 | −7.5 |
| Hesperetin:UNK1:H2–SPK2:GLN625:O | 130.888 | 1.94553 | |
| Hesperetin:UNK1:H3–SPK2:ALA431:O | 105.734 | 2.29542 | |
| SPK2:ASP110:HN–Hesperetin:UNK1:O17 | 152.041 | 2.32426 | −7.4 |
| Hesperetin:UNK1:H1–SPK2:CYS100:O | 138.862 | 2.43301 | |
| Hesperetin:UNK1:H2–SPK2:ARG373:O | 120.303 | 2.05193 | −7.3 |
| Hesperetin:UNK1:H3–SPK2:ASP342:OD1 | 147.571 | 2.13992 | |
| SPK2:GLY248:HN–Hesperetin:UNK1:O19 | 134.859 | 2.21696 | −7.1 |
| Hesperetin:UNK1:H1–SPK2:CYS276:O | 89.142 | 2.41654 | |
| Hesperetin:UNK1:H1–SPK2:THR641:OG1 | 135.818 | 2.17768 | −7.0 |

unfavourable total electrostatic contributions (EEL + EGB). The favourable binding free energy of −28.18 and −17.46 kcal/mol were for 3GPG-hesperetin and 3TRK-hesperetin, respectively. The estimated binding free energy calculated is in agreement with the docking study.

## DISCUSSION

Since the initial Tanzania outbreak in 1952 CHIKV epidemics have been recorded in several parts of the world with different climate systems. The spread of CHIKV to countries outside the tropical regions has been attributed to the single amino acid alteration from alanine to valine (A226V) in the E1 gene (*Tsetsarkin et al., 2007*). To date, approved therapeutic options and vaccines for CHIKV infections are lacking, though cases of CHIKV infected patients are constantly on the rise globally. Mechanistic studies in the past have shown anti-CHIKV effect of several compounds as a result of their interactions with different viral proteins, hence inhibiting further CHIKV replication (*Delogu et al., 2011*; *Kaur et al., 2013*; *Lani et al., 2015*). Furthermore, computational approaches have been applied by researchers in identifying potential interactions between drug candidates with specific CHIKV proteins in an effort to discover a novel anti-Chikungunya treatment (*Agarwal, Asthana & Bissoyi, 2015*; *Nguyen, Yu & Keller, 2014*; *Singh et al., 2012*). In this study, we aim to investigate the potential inhibitory effects of hesperetin, a flavanone, on different CHIVK non-structural proteins in addition to one of the cellular factors vital for CHIKV replication, SPK2.
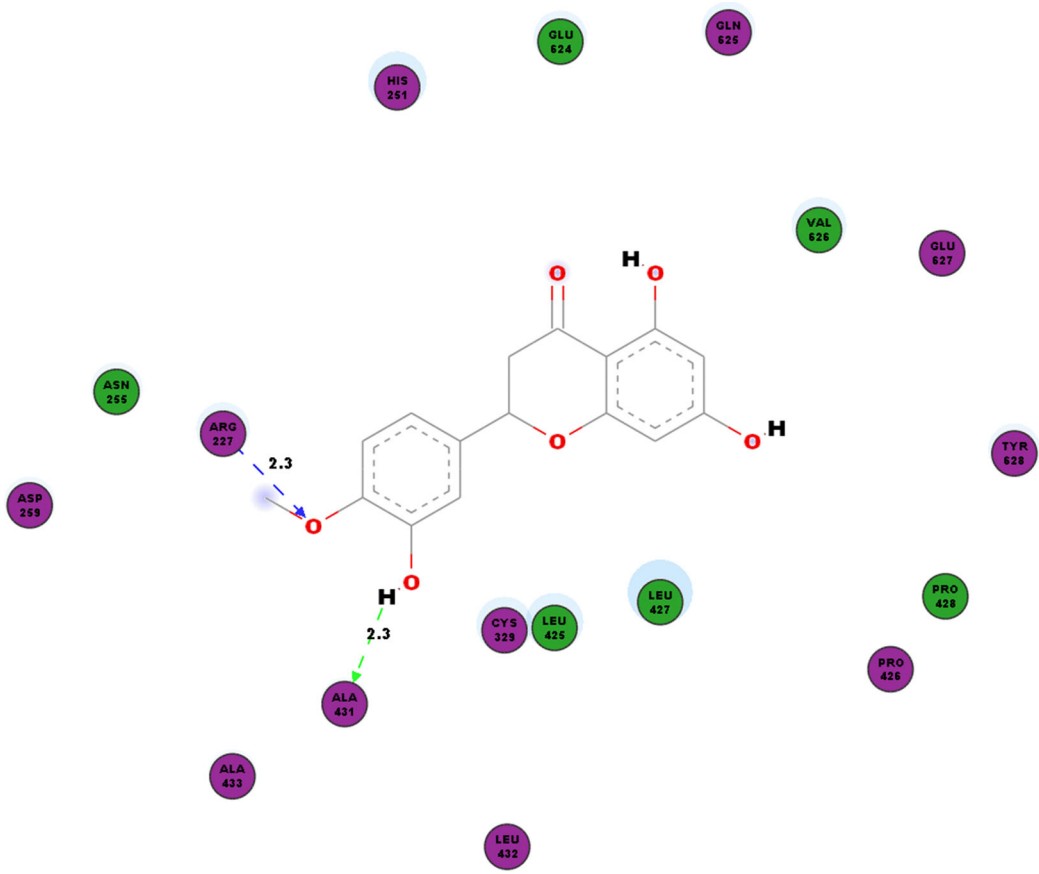

**Figure 6 2D diagram of interaction between hesperetin and SPK2.** The diagram demonstrates the ligand-receptor interactions and close amino acid residues found in the binding pocket.

**Table 9 Lipinski's values of the hesperetin molecule designed for this study and that of the available data from PubChem.** A compound with drug-likeness properties should possess molecular weight less than 500 g/mol, Log P less than 5, less than 5 H-bond donors, less than 10 H-bond acceptors and molar refractivity in the range of 40–130.

| Criteria | From the present study | Extracted from PubChem |
|---|---|---|
| Molecular weight (g/mol) | 302.00 | 302.28 |
| H-bond donors | 3 | 3 |
| H-bond acceptors | 6 | 6 |
| Log-P | 2.5 | 2.4 |
| Molar refractivity | 76.75 | 76.93 |

Our findings showed that hesperetin possesses interactions with all four CHIKV non-structural proteins as well as SPK2 at different binding affinities. This is consistent with the inhibitory efficacy of hesperetin during the virus intracellular replication observed during our previous in vitro screening (*Ahmadi et al., 2016*). Hesperetin has the highest tendency to bind with 3GPG with the affinity of −8.5 kcal/mol, followed by both nsp4 and SPK2 at −7.7 kcal/mol, nsP1 with the binding strength of −7.6 kcal/mol and finally 3TRK bound at −6.9 kcal/mol. Hence, CHIKV's nsP3 or 3GPG has the highest

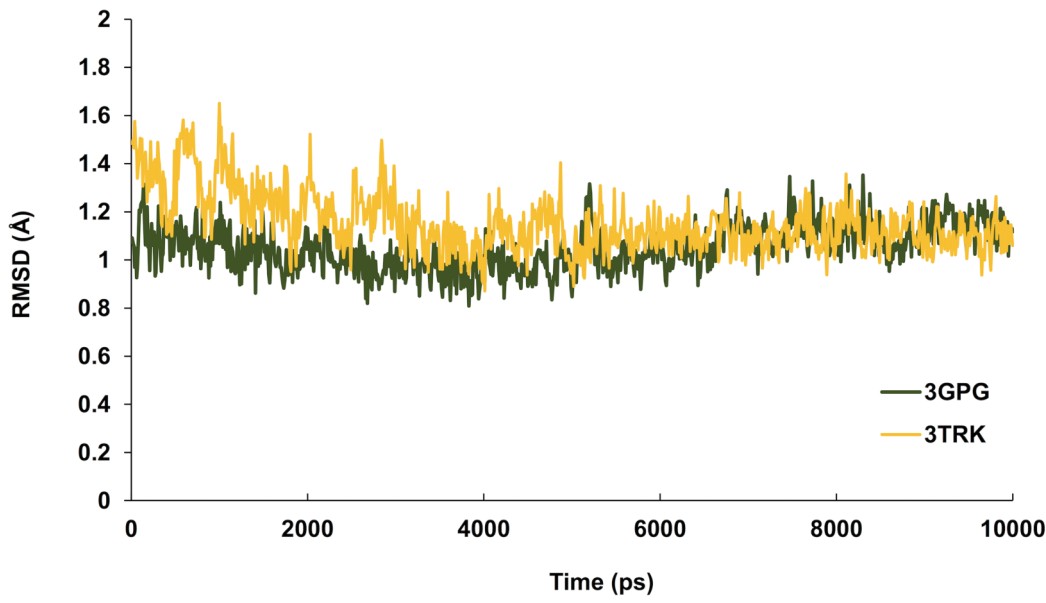

**Figure 7 Time series of RMSD from the minimized starting structure calculated using the backbone atoms of the protein.** RMSD computed with respect to the minimized starting structure is less than 1.8 Å, hence showing that the simulated systems are well equilibrated and do not deviate much from the initial starting structure.

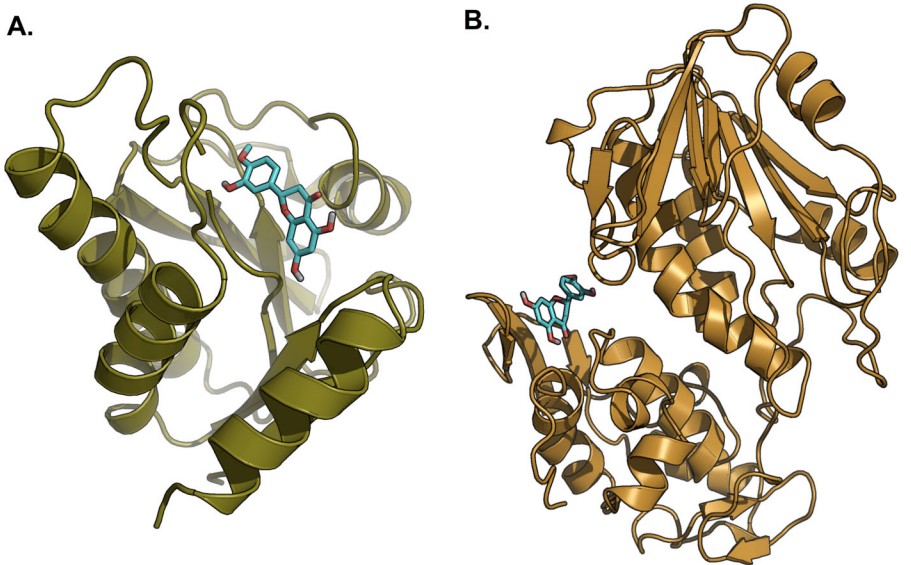

**Figure 8 Lowest energy structure of ligand-receptor complexes.** (A) 3GPG and (B) 3TRK in complexed with hesperetin (shown in stick representation, only polar hydrogen is shown). Both complexes remained stable throughout the 10 ns simulations.

potential to be the target protein for hesperetin's inhibitory activity on the virus intracellular replication, though other proteins under study should not be excluded of the possibilities. Interestingly, CHIKV nsP3 protein plays a vital role in the early stages of viral genome replication involving synthesis of viral negative-strand and subgenomic RNAs (*Lani et al., 2015*). Previous study has also found that mutations resulting in

**Table 10 Relative binding free energies of complexes estimated using MM-GBSA.**

| Complex | EEL | vdW | EGB | ESURF | $\Delta E_{binding}$ |
|---|---|---|---|---|---|
| 3GPG + hesperetin | −10.26 | −36.62 | 23.47 | −4.76 | −28.18 |
| 3TRK + hesperetin | −4.13 | −28.32 | 18.61 | −3.61 | −17.46 |

Notes:

The EEL and vdW represent the electrostatic and van der Waals contributions from MM, respectively. EGB stands for GB electrostatic contribution to the solvation free energy, and ESURF is the nonpolar contribution to the solvation free energy.

$\Delta E_{binding}$ (in kcal mol$^{-1}$, binding energy neglecting the contribution of entropy) is the final estimated binding free energy calculated from the terms above.

alteration of the protein's amphiphysin SH3 binding motif and downregulation of Amph2 expression have been found to diminish in vitro viral RNA replication (*Neuvonen et al., 2011*). Inhibition of both the nsP3 and nsP4 proteins could also block CHIKV replication via the heat shock protein, HSP-90 pathway (*Rathore et al., 2014*). Besides the virus polymerase nsP4 which also modulate host immune response towards CHIKV infections via the PKR-like ER resident kinase (PERK) pathway (*Rathore, Ng & Vasudevan, 2013*), the host factor SPK2 was also ranked second by the AutoDock Vina 1.5.6 based on their binding affinities. NsP3-mediated SPK2 re-localisation to a puncta in the cytoplasm following an infection suggested the involvement of SPK2 in CHIKV replication. Close association of SPK2 with host proteins for mRNA processing and gene expression further link SPK2 with CHIKV replication process (*Reid et al., 2015*). Our findings suggest that inhibitory effect of hesperetin could be contributed by both direct activity on the virus proteins as well as cellular factors vital for the virus replication.

We have also identified the different types of non-covalent bonds between the ligand and the receptor proteins, including the hydrogen bonds, pi-pi, pi-cation and pi-sigma interactions. Hydrogen bond is a dipole-dipole interaction involving the transfer of a proton from one molecule to the electronegative recipient atom on another molecule. Hydrogen bond varies in strength based on the distance of the hydrogen bond, whereby as described by Szatylowicz, hydrogen bonds are classified as strong, moderate and weak if they fall into the ranges of 1.2–1.5, 1.5–2.2 and > 2.2 Å, respectively (*Szatyłowicz, 2008*). None of the hydrogen bonds formed between hesperetin and all five proteins in this study fall under the strong category. Instead, hesperetin formed moderately strong hydrogen bonds with relatively strong binding affinities with ASP310, GLY311, ASP36, ARG71 and LEU135 of nsP1, TYR142, LEU108, SER110, ARG144, GLU17, THR122 and ASN72 of 3GPG, GLN364, ALA104, ALA103, ARG110, LEU135, ASP479 and THR437 of nsP4, and finally ARG373, GLN625 and ASP342 of SPK2. Though strong hydrogen bonds were also observed between hesperetin and 3TRK at ASP1246 and TRP1084, their binding affinities were the lowest relative to the other four proteins studied.

Another type of non-covalent interaction investigated in this experiment are the pi interactions, which involve the electron-rich pi-system with another pi-system, anion, cation or other molecules. Among all the pi-interactions investigated in this study, the pi-sigma bonds have the strongest binding strength due to the larger overlaps of its s-orbitals (*Harmony, 1990*; *Sinanoglu & Wiberg, 1963*). From our findings, we observed four

pi-sigma interactions between hesperetin and 3TRK at PHE1225, 3GPG at GLY30 and VAL113 as well as ARG70 of nsP4. We have also documented the strong electrostatic forces between a polarised ligand and a receptor's binding site, as well as the van der Waals interaction energies. Van der Waals interactions, which are generally weaker as compared with the other forces, are formed when the cationic nucleus of a molecule is attracted to the electron cloud of another molecule, or vice versa. It results from transient alterations of polarisation in the nucleus affecting the surrounding dipole-dipole interactions and the nearby electron distribution. The weak van der Waals are complemented with the presence of large numbers of bindings arising from good electrostatic and steric interactions between both molecules (*Copeland, 2013*).

The Lipinski's Rule of Five, also known as the Pfizer's Rule of Five by Christopher A. Lipinski explains the basic criteria which one compound needs to possess to exhibit druglikeness properties. Compounds which fulfil the following requirements are most likely to be membrane permeable and have high bioavailability following oral administration–molecular weight less than 500 g/mol, Log P less than five, less than 5 H-bond donors, less than 10 H-bond acceptors and molar refractivity in the range of 40–130 (*Lipinski et al., 2012*). As shown in Table 9, the hesperetin molecule used in our study matches all five of the Lipinski's Rule of Five–molecular weight of 302 g/mol, Log P of 2.5, 3 H-bond donors, 6 H-bond acceptors and molar refractivity of 76.75. We compared our findings with the available data from PubChem to ensure that the ligand designed is reliable for our docking experiment. The results suggest that hesperetin has the necessary properties to be developed into an effective oral therapeutic for CHIKV infections. Furthermore, the non-structural proteins (3GPG and 3TRK) in complexed with hesperetin remained stable throughout the 10 ns MD simulations. In all, these findings are of utmost importance for determining whether hesperetin has the potential to be developed into an effective antiviral drug against CHIKV infections as well as its highest possible target protein.

## CONCLUSION

Hesperetin possesses interactions with all four CHIKV non-structural proteins in addition to SPK2 which plays a role in the virus replication cycle. These findings enhance our understandings on the possible underlying inhibitory mechanism of CHIKV replication, hence allowing further studies on these target proteins for the development of novel anti-CHIKV drug.

### Funding

This study was supported by the Ministry of Higher Education, Malaysia High Impact Research (HIR) MOHE Grant (H20001-E000087) and the University of Malaya Research Grant (RG356-15AFR). The Data Intensive Computing Centre (DICC), University of Malaya provided computational resources. The funders had no role in study design, data collection and analysis, decision to publish, or preparation of the manuscript.

## Grant Disclosures

The following grant information was disclosed by the authors:

Ministry of Higher Education, Malaysia High Impact Research (HIR) MOHE Grant: H20001-E000087.

The University of Malaya Research Grant: RG356-15AFR.

## Competing Interests

The authors declare that they have no competing interests.

## Author Contributions

- Adrian Oo conceived and designed the experiments, performed the experiments, analyzed the data, wrote the paper, prepared figures and/or tables, reviewed drafts of the paper.
- Pouya Hassandarvish performed the experiments, reviewed drafts of the paper.
- Sek Peng Chin performed the experiments, analyzed the data, prepared figures and/or tables.
- Vannajan Sanghiran Lee performed the experiments, analyzed the data, contributed reagents/materials/analysis tools, prepared figures and/or tables, reviewed drafts of the paper.
- Sazaly Abu Bakar conceived and designed the experiments, contributed reagents/materials/analysis tools, reviewed drafts of the paper.
- Keivan Zandi conceived and designed the experiments, analyzed the data, contributed reagents/materials/analysis tools, reviewed drafts of the paper.

## Data Deposition

The raw data has been supplied as Supplemental Dataset Files.

## Supplemental Information

Supplemental information for this article can be found online at http://dx.doi.org/10.7717/peerj.2602#supplemental-information.

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
