# Peer review of "In silico study on anti-Chikungunya virus activity of hesperetin"

_PeerJ, doi:10.7717/peerj.2602_

## Round 0.1 · original submission · Minor Revisions

The reviewers have provided positive comments along with some useful suggestions for changes that will improve your paper. Please respond to the comments with a good faith effort to address all of these issues before returning your revised manuscript for consideration.

·

Basic reporting

Results of the work in consideration shows significant effects of hesperetin on Various NSP domains. The discussion part, rightly pointed out the future study in the field for deciphering the molecular mechanisms. A new drug molecule based on stuctural properties hesperetin can developed for treatment of Chikungunya virus.


The paper meet the appropriate standard for the publication(with some correction and experiments)

Experimental design

Comment 1: To start with, there are a lot of sentences are incomplete and abrupt reflecting the casual approach adopted by authors in framing the research paper.
Comment 2: The experiment carried out using 3-dimensional structures of CHIKV nsP2 and nsP3 were retrieved from the Protein Data Bank (PDB), whereas nsP1, nsP4 and the cellular factor sphingosine kinase 2 (SPK2) were designed using the Iterative Threading.
Also molecular docking on hesperetin against all four CHIKV non-structural proteins and SPK2. Proteins preparation and subsequent molecular docking with the ligand were performed using appropriate software.
I suggest author to carryout Molecular dynamic simulation study of nsP domains and hesperetin complex at least for 10ns to find out the stability of complex.
For reference: Agarwal, T., Somya Asthana, and A. Bissoyi. "Molecular modeling and docking study to elucidate novel chikungunya virus nsP2 protease inhibitors." Indian journal of pharmaceutical sciences 77, no. 4 (2015): 453.

Validity of the findings

No Comments

Additional comments

No Comments

·

Basic reporting

First and foremost, the authors have to be acknowledged and appreciated for identifying and designing an inhibitor Hesperetin that could be suitable enough to curb CHIKV. Since decades there hasn’t been a suitable inhibitor for CHIKV but this study has sought us with a way and a solution towards Hesperetin through computational approaches. In this study, four non-structural proteins responsible in facilitating the onset and progression of virality of CHIKV were thoroughly studied and the mechanism of action was laid out by precisely analyzing the variety of interactions scale and their occurrence pattern. The best target was then chosen on the basis of interaction pattern and the binding affinity score.

Experimental design

In this study, though proteins were modeled due to the absence of structural substantiation using I-TASSER template information of the same was not mentioned anywhere in the text. Before structure generation usually sequences are aligned in order to get an insight on the conserved residues and the active site information which can be picked in generating the grid surrounding which the ligand can be docked. This sort of methodology usually provides information on so many aspects and guides us to follow the work properly. In the I-TASSER five models are generated for every structure out of which based on C-score, the best model is selected for further studies and subjected to structure validation and molecular dynamics simulation to ensure the structural stability and conformational variability after which molecular docking is performed but the authors haven’t performed such checks on the experimental design. Following the molecular docking, Molecular dynamics simulation of the best target and the compound is performed to check for the same reasons.

Validity of the findings

The findings projected in this study are reasonable enough to deduce on an appropriate target based on interactions and binding affinity but there are other checks which should be performed like apart from the docking, Simulations of the complex could ensure the influence upon ligand binding with the target and such motions can be monitored with utmost accuracy and a detailed enumeration could provide much insights.

Additional comments

This concept is wonderfully perceived through these analyses.
• Generating the model(s) of nsp1, ns2 and SPK2.
• Selection of already existing structures from PDB nsp2 (3TRK) and nsp3 (3GPG)
• Preparation of target structures and Hesperetin
• Docking of Hesperetin against each of the target structures.
• Selection of best target nsp3 for Hesperetin based on binding affinity, non-covalent interactions and special interactions
In addition, I would like to mention that the writing part was good enough but I feel that the use of phrases could be simplified for easy understanding. The study is well designed and the results are supported by their calculations /analyses. However, I suggest major revisions.
The following are comments and critiques that should be addressed in a revision of the manuscript:
1. Under Results section in 4th line, what does different binding conformation(s) correspond to? I suggest “conformation” means the overall motion of the protein when it comes to ligand docking movements it has to be represented by poses.
2. In the subsequent para, lines (217-219) the residues listed are found surrounded in the binding cavity how they can represent close interactions when they don’t form any sort of interactions eventhough there are some residues not found in close vicinity to the ligand and will not fit the criteria of forming interaction. Please check and modify accordingly.
3. Line 223, in table: 7 pi-interactions corresponding to nsp2 such as Tyr1077 which is found in the text whereas in the table it has been found as Tyr1079. Check and modify accordingly.
4. In the subsequent last line it has been labeled as Leu1206 in Fig.3 whereas in the text it has been found as Leu1205 and in the very figure interactions with MSE, a modified resdiue doesn’t have any sort of prominent effect. Check and correct it.
5. In the following para, line 230 Ser110 found to be form interaction whereas the other two (Gly112 and Cys143) listed as above are just found to be available in the binding pocket and in any waydoesn’t form any interactions with the compound.
6. Therefore, all the residues listed for all the targeted proteins are the ones present in the binding pocket which might be liable to form interaction but didn’t due to the distance criteria. Modify the sentences accordingly.
7. Line237, Tyr22 and TYR185 represent both pi-sigma and pi-cation. Check and correct accordingly.
8. Lines244-245 either mention all the residues from the binding pocket some are missing. Follow the consistency everywhere
9. All the figures have to improved as they obscure the readers of understanding the clear picture from the text and the correlated illustration.
Cite the following reference in your study and could be immensely suited and useful
"Singh, K. D., Kirubakaran, P., Nagarajan, S., Sakkiah, S., Muthusamy, K., Velmurugan, D & Jeyakanthan, J. (2011). Homology modeling, molecular dynamics, e-pharmacophore mapping and docking study of chikungunya virus nsP2 protease. Journal of Molecular Modeling, 18, 39–51. DOI: 10.1007/s00894-011-1018-3"
Overall, I would recommend a more careful reading of the manuscript for clarity and brevity to eliminate any confusion.

---

## Round 0.2 · accepted · Accept

Thank you for submitting this fine piece of work to PeerJ.